# The Role of Isoflavones in Type 2 Diabetes Prevention and Treatment—A Narrative Review

**DOI:** 10.3390/ijms22010218

**Published:** 2020-12-28

**Authors:** Alina Kuryłowicz

**Affiliations:** Department of Human Epigenetics, Mossakowski Medical Research Centre, Polish Academy of Sciences, 5 Pawinskiego Street, 02-106 Warsaw, Poland; akurylowicz@imdik.pan.pl; Tel.: +48-226-086-591; Fax: +48-226-086-410

**Keywords:** isoflavones, genistein, daidzein, glyctin, formononetin, biochanin A, type 2 diabetes

## Abstract

Given the growing number of type 2 diabetic individuals and the substantial social and financial costs associated with diabetes management, every effort should be made to improve its prevention and treatment methods. There is an ongoing search for natural dietary compounds that could be used for this purpose. This narrative review focuses on the therapeutic potential of isoflavones in diabetes prevention and treatment. This review summarizes (i) the molecular mechanisms of isoflavones action that are critical to their anti-diabetic properties; (ii) preclinical (in vitro and in vivo) studies evaluating the influence of isoflavones on the function of key organs involved in the pathogenesis of diabetes; and (iii) epidemiological studies and clinical trials that assessed the effectiveness of isoflavones in the prevention and treatment of type 2 diabetes in humans. Apart from discussing the effects of isoflavones on the function of organs “classically” associated with the pathogenesis of diabetes (pancreas, liver, muscles, and adipose tissue), the impact of these compounds on other organs that contribute to the glucose homeostasis (gastrointestinal tract, kidneys, and brain) is also reviewed.

## 1. Introduction

According to data from the World Health Organization and International Diabetes Foundation, one tenth of adults around the world suffer from diabetes, and among people over 65, this rises to one fifth. Type 2 diabetes (T2D) accounts for approximately 90% of all diabetes cases and even though the number of patients diagnosed with this disease is growing every year it is still assumed that half of cases remain undiagnosed. Moreover, approximately 400 million people worldwide are at increased risk of developing T2D [1].

Responsible for over 1.5 million deaths a year and the leading cause of blindness, chronic renal failure, cardiovascular diseases, and lower limb amputation, T2D is a serious health, social, and economic problem [2]. Lifestyle interventions based on a healthy diet, regular physical activity, maintaining normal body weight, and avoiding tobacco smoking have a proven role in preventing or delaying the onset of T2D. Therefore, in parallel with the development of new groups of drugs registered for diabetes treatment, there is a continual search for natural ingredients and dietary supplements that could be useful for preventing or modifying the course of diabetes.

The maintenance of glucose homeostasis depends on the pancreatic β-cells’ normal insulin secretion and the normal sensitivity of tissues (mainly liver and muscles) to insulin. However, according to the ominous octet concept, accelerated lipolysis in adipose tissue, disturbed incretin secretion, hyperglucagonemia caused by α-cell dysfunction, increased glucose reabsorption in kidneys, and abnormal brain response to insulin all contribute to the development of hyperglycemia [3]. To obtain normoglycemia, the compound of interest should exert a pleiotropic effect on the number of cells and tissues. This narrative review discusses whether isoflavones—plant-derived dietary compounds—influence the function of key organs involved in the pathogenesis of diabetes and how these effects may translate to clinical practice. Apart from discussing the effects of these compounds on the function of organs “classically” associated with the pathogenesis of T2D (pancreas, liver, muscles, and adipose tissue), their impact on other organs that are crucial for glucose homeostasis maintenance (gastrointestinal tract, kidneys, and brain) is also reviewed. Even though this review does not meet the rigorous criteria of a systematic one and may include an element of selection bias, it describes and summarizes several concepts regarding the mechanisms of isoflavones action that could be applied in T2D prevention and treatment.

## 2. Methods

To summarize findings regarding the effects of isoflavones on the pathogenesis, incidence, and course of T2D, a literature search was performed in the PubMed database. Articles used for this review addressed the impact of isoflavones on the function of key organs involved in the pathogenesis of T2D as well as its incidence and course in three experimental setups (cell, animal, and human studies). A systematic literature search was conducted from 1991 to 2020 using the following terms “isoflavones” OR “genistein” OR “daidzein” OR “glyctin” OR “formononetin” OR “biochanin A” combined with “type 2 diabetes mellitus” OR “pancreatic islets” OR “pancreatic β-cell” OR “pancreatic α-cell” OR “liver” OR “muscle” OR “adipose tissue” OR “kidney” OR “gastrointestinal tract” OR “intestine” OR “brain” OR “appetite”. Only articles published in peer-reviewed scientific journals were included in the analysis. Of the 754 identified papers, after excluding titles and abstracts for not meeting the inclusion criteria, 143 unique articles were identified that met the review criteria, which comprised 43 human studies and 100 animal and cell line studies.

## 3. Isoflavones

### 3.1. Classification and Metabolism of Isoflavones

Isoflavones, together with coumestans and prenylflavonoids, belong to a group of flavonoid phytoestrogens that naturally occur in nonsteroidal phenolic plant compounds. They can be found in the Leguminosae family as non-active hydrophilic glycosides (e.g., daidzin, genistin, and glyctin in soybean) and 4’-methylated lipophilic derivatives (e.g., formononetin and biochanin A in red clover) [4].

In the gastrointestinal tract, non-active isoflavone glycosides are hydrolyzed into bioactive aglycones by brush border membrane and bacterial β-glucosidases (e.g., daidzin to daidzein, genistin to genistein, and glyctin into glycytein), which are next absorbed across the intestinal epithelium. After the absorption, they are metabolized to β-glucuronides and sulfate esters in the intestinal mucosa cells. These metabolites can be found in the plasma and excreted in the bile and are then subsequently deconjugated in the intestine’s distal part, which allows their reabsorption [5].

In general, isoflavones and their metabolites have a short half-life (about 9 h for daidzein and 7 h for genistein) and are eliminated with urine. However, some aglycones undergo further transformation by intestinal microflora. For instance, genistein is converted to dihydrogenistein and subsequently *p*-ethyl-phenol and 6-hydroxy-*O*-desmethylangolensin. In turn, daidzein can be metabolized to dihydrodaidzein, which is converted to equol, a metabolite with a strong estrogenic and antioxidant activity. Urinary concentrations of equol are commonly used for a screening of the soy protein dietary intake. However, it should be stressed that equol production depends on the genetic predisposition and composition of intestinal microflora, and about half of the adult population do not excrete equol in the urine [6].

### 3.2. Mechanisms of Isoflavones Action

The pleiotropic effects of isoflavones on key organs involved in the pathogenesis of diabetes function result from their ability to modify several cellular pathways by triggering various mechanisms. A detailed description of the isoflavone action mechanisms is not the subject of this review and may be found elsewhere [7,8]. Here, I briefly describe those mechanisms that may be important for regulating organs and tissues, crucial for the maintenance of glucose homeostasis. In general, isoflavones exert their influence on cells via the modulation of nuclear receptor activity and non-receptor signaling.

As phytoestrogens, isoflavones show structural similarity to 17-β-estradiol (E2); therefore, they have an affinity to estrogen receptors (ER) α and β. These receptors exist in multiple splice variants and exhibit tissue specificity in expression and function. ERs act as transcription factors that, upon ligand binding, undergo conformational changes, homodimerization, nuclear translocation, and finally interaction with either estrogen response elements (ERE) in the target gene promoters or other regulatory elements (e.g., transcription factors) bound to DNA. Subsequently, after the recruitment of coactivators and interaction with other basal transcription complex components, isoflavones regulate transcription of target genes [9]. The final effect (agonistic or an antagonistic) of the particular isoflavone exerted by binding to ER is determined by several factors including (i) its affinity towards ERs isoforms, (ii) the mutual proportion of the ER isoforms in a given tissue, (iii) the local availability of ER coactivators and corepressors, and (iv) the local concentration of isoflavone and its relation to the local level of endogenous estrogens [10]. Some isoflavones (e.g., genistein and equol) can activate membrane-associated forms of ERα coupled with G proteins, growth factor receptors, tyrosine kinases (Src), etc. In this case, receptor binding activates protein kinases that phosphorylate other transcription factors and promote their nuclear translocation and transcriptional action [11].

Peroxisome proliferator-activated receptors (PPARs) represent another group of nuclear receptors activated by isoflavones (e.g., genistein, daidzein, formononetin, and biochanin A). This mechanism of isoflavone action is of particular interest in the context of diabetes, since the activation of PPARγ plays a central role in the regulation of insulin sensitivity and blood glucose homeostasis [12]. Isoflavones also have the potential to activate PPARα, a chief regulator of genes involved in fatty acid β-oxidation, and PPARδ, a fatty acid sensor that regulates a variety of genes implicated in lipid metabolism [13].

Both formononetin and biochanin A were found to activate the aryl hydrocarbon receptor (AhR), a ligand-activated transcription factor which dimerizes with aryl hydrocarbon receptor nuclear translocator (ARNT), binds to xenobiotics response element (XREs) and modulates the expression of downstream genes. Apart from the regulation of xenobiotic metabolism, AhR mediates in pathways related to cell cycle and apoptosis and plays a role in liver fibrosis development [14]. In turn, genistein and daidzein activate nuclear respiratory factors (Nrfs), which are transcription factors essential for the regulation of antioxidative enzyme expression cellular defense against the toxicity of reactive oxygen species (ROS) [15].

In vitro studies showed that isoflavones act in mechanisms independent of nuclear receptors. These include the inhibition of protein tyrosine kinases (e.g., extracellular signal-regulated kinases 1 and 2—ERK1/2), whereby they regulate several cellular functions such as cell proliferation and differentiation [16]. Isoflavones also exhibit antioxidant activity, which is independent of their estrogenic properties [17]. By reduction of reactive oxygen species (ROS) levels and induction of the antioxidant enzymes, isoflavones may activate the adenosine monophosphate-activated protein kinase (AMPK) that plays a central role in the regulation of energy metabolism [18]. In turn, the inhibition of cyclooxygenase 1 (COX-1) activity and thromboxane A_2_ (TXA_2_) synthesis, as well as nuclear factor κB (NF-κB) pathway, contributes to the anti-inflammatory properties of isoflavones [19,20]. Interestingly, the inhibition of NF-κB signaling by isoflavones may occur directly or via the activation of sirtuins (SIRT), which are NAD-dependent deacetylases that control the expression of other genes by modification of histones and transcription factors [21]. Some isoflavones inhibit DNA topoisomerases I and II and ribosomal S6 kinase [22].

Among the non-receptor mechanisms of action of isoflavones, the ability to modify the epigenome should also be mentioned. In the in vitro model, genistein, daidzein, and equol were found to decrease methylation of several tumor suppressor genes in cancer cell lines through the inhibition of DNA methyltransferase (DNMT) activity [23]. However, the influence of isoflavones on methylation status may depend on the experimental model, since in animal studies the administration of genistein resulted in increased DNA methylation levels in the liver and muscle tissues that were associated with increased insulin sensitivity [24]. In clinical trials, treatment with genistein, daidzein, and glycitein caused hypermethylation of various genes encoding tumor suppressors [25].

The role of isoflavones in the regulation of microRNA (miRNA) in diabetes and its complications is less studied. The antioxidant activity of genistein associated with preventing atherosclerosis also depends on the inhibition of miR-34a and subsequent upregulation of SIRT1. Its anti-inflammatory properties—decreased expression of E-selectin, P-selectin, monocyte chemotactic protein-1 (MCP-1), interleukin-8 (IL-8), vascular adhesion molecule-1 (VCAM-1), and intercellular adhesion molecule-1 (ICAM1)—depend on the inhibition of miR-155, while its ability to improve insulin sensitivity in muscles and to inhibit lipolysis in adipose tissue depends on the inhibition of miR-222 [21,26,27,28]. It should be noted that some of the mechanisms described here have been demonstrated in vitro using concentrations of isoflavones exceeding those that naturally occur, so findings require confirmation using in vivo tests.

Mechanisms of isoflavones action relevant to the pathogenesis of T2D are summarized in Figure 1.

## 4. Influence of Isoflavones on the Function of Organs Critical for Diabetes Development and Progression

Through the mechanisms described in Section 3, isoflavones may regulate the function of organs essential for glycemic control and therefore modify the development and course of diabetes.

### 4.1. Pancreatic Islets

During the early stages of T2D, the pancreas increases insulin secretion to counteract insulin resistance. Even though impaired glucose tolerance (IGT) occurs when the pancreas loses from 60% to 85% of the total insulin secretory capacity, the onset of β-cells failure occurs earlier. The decrease in insulin secretion is caused by the dysfunction of β-cells and a progressive decrease in their number.

In humans, islets constitute approximately 1–2% of total pancreas mass and up to 70% of its cells are β-cells. The total number of β-cells results from several processes including neogenesis (differentiation of precursor cells into β-cells), transdifferentiation (differentiation of other cell types into β-cells), a proliferation of pre-existing β-cells, and apoptosis [29]. The prediabetic individual’s final progression to overt diabetes is synonymous with a decline in insulin secretion and is accompanied by the loss of functional β-cell mass [30]. Maintaining an adequate β-cell mass by inhibiting β-cell death represents a possible strategy of prevention and treatment of diabetes. Isoflavones have been shown to support β-cell survival and function in several in vitro and in vivo studies [20,21,22,23,24,25,26,27,28,29,30,31,32,33,34,35,36,37,38,39,40,41,42,43,44].

Genistein can prevent the pro-inflammatory cytokines-mediated reduction of glucose-stimulated insulin secretion (GSIS) and stimulates proliferation in rat insulinoma (RIN) cells. These effects of genistein were attributed to suppression of the NF-κB and ERK1/2 pathways [20]. In turn, genistein’s ability to counteract high-glucose-mediated cell damage and DNA fragmentation is mediated by ERs [31]. However, it should be stressed that the influence of genistein on β-cells apoptosis may depend on its concentration. For instance, at a concentration of 100 μM, genistein increased apoptosis in rat and human islets, while at a concentration of 25 μM, it exerted an anti-apoptotic effect [32]. Both genistein and biochanin A, through the activation of AMPK, enhanced cell replication and increased cyclin D1, a major cell cycle regulator for β-cell proliferation in rat insulinoma (INS-1) cells as well as in human islet β-cells [33].

Genistein and daidzein activate AMPK and calcineurin signaling pathways, potentiating GSIS by β-cells. However, similar to the case of regulation of apoptosis, this effect is concentration-dependent [33,34,35,36]. Doses of isoflavones applied in these experiments were supraphysiological, which may raise concerns that the observed phenomena would not occur in living organisms.

The anti-diabetic effect of isoflavones resulting from β-cells protection was confirmed in several animal models of diabetes. At doses representing a typical daily intake of a soy-rich diet in humans, genistein protected β-cells and maintained insulin production in rats and mice treated with streptozotocin (STZ, a toxic alkylating agent exclusively transported into β-cells) and alloxan (a toxic glucose analog that destroys β-cells). This effect was obtained due to genistein’s ability to enhance β-cells proliferation and reduce apoptosis [33,37,38]. Importantly, through the protection of β-cells and stimulation of insulin production, genistein is also able to reduce hyperglycemia in diet-induced obesity models (e.g., in STZ treated C57BL/6 mice on a high-fat diet—HFD), ovariectomized rats, and in genetically obese diabetic (db/db) mice, which share the metabolic characteristics of human T2D manifested with insulin resistance and reduced β-cell mass and function [39,40,41].

Oxidative stress is a major cause of β-cell death, because the antioxidative capacity of β-cells is relatively low. Equol S-enantiomer counteracts alloxan-induced oxidative stress in INS-1 cells with higher activity than its precursor daidzein. The underlying molecular mechanisms of this action include AMPK activation, phosphorylation of the cAMP-response element-binding protein, and induction of cAMP-response element-mediated transcription as well as, decrease in activity of protein kinase A (PKA) and the increase in activity of protein phosphatase (PP2A) [42,43]. S-equol treatment increases the number of Ki67-positive proliferating β-cells and decreases the number of apoptotic β-cells, providing resistance to hyperglycemia after alloxan treatment in mice and in Zucker diabetic fatty rats [42,43]. Formononetin reversed alloxan-induced hyperglycemia in mice through inhibition of β-cell apoptosis and promotion of cell regeneration, insulin secretion, and by increasing hepatic glycogen synthesis and hepatic glycolysis [44]. Formononetin’s ability to protect β-cells from necro-degeneration and atrophy is mediated by an increase of SIRT1 expression in pancreatic islets and subsequent inhibition of NF-κB signaling [45,46]. Additionally, biochanin A was found to have antihyperglycemic activity, since, at doses of 10 and 20 mg/kg body weight, it decreased plasma glucose and glycosylated haemoglobin levels and increased the plasma insulin level in STZ-induced diabetic rats [47,48]. However, higher (40 mg/kg) doses of biochanin A significantly reduced glucose tolerance in this experimental model [48].

Plasma glucagon levels are constantly elevated in T2D patients and contribute to hyperglycemia by stimulating hepatic gluconeogenesis [3]. The influence of isoflavones on pancreatic α-cells has not been studied extensively. Nevertheless, a three week administration of daidzein to geese resulted in a significant increase in insulin and glucagon blood levels compared to control animals [49]. Similar effects were obtained after a five week daidzein treatment in broilers and db/db mice, where administration of both genistein and daidzein reduced glucagon secretion without any significant impact on the β-cells function [42,50]. Moreover, genistein reduced pancreatic polypeptide levels in diabetic leptin-deficient ob/ob mice [51]. However, administration of formononetin had no influence on glucagon secretion in alloxan-induced diabetic mice [44].

### 4.2. Liver

The liver of a diabetic patient is a major site of insulin resistance that manifests itself by an overproduction of glucose during the basal state, despite fasting hyperinsulinemia and hyperglycemia and impaired suppression of hepatic glucose production in response to postprandial insulin secretion. Approximately 70–80% of obese and diabetic patients suffer from non-alcoholic fatty liver disease (NAFLD) associated with more severe insulin resistance, hyperinsulinemia, and dyslipidemia. The pathogenesis of the NAFLD-associated insulin resistance is complex. Adipose tissue-derived cytokines (e.g., tumor necrosis factor (TNF) α, IL-1β, and IL-6) play a significant role in this process and are activated by pro-inflammatory pathways and the inhibition of insulin receptor signaling in the hepatocyte. In addition, excessive hepatic lipid accumulation promotes liver infiltration by macrophages, natural killer cells, and T-cells, which are the primary source of pro-inflammatory cytokines that contribute to the development of insulin resistance and progression of NAFLD to the non-alcoholic steatohepatitis (NASH) [52].

Genistein and daidzein supplementation decreases gluconeogenic enzyme activity in the liver (namely glucose-6-phosphatase and phosphoenolpyruvate carboxykinase) and increases the activity of the malic enzyme and glucose-6-phosphate dehydrogenase in non-obese diabetic (NOD) mice. These changes occur in parallel to the increased hepatic glycogen contents and lower blood glucose levels in treated animals. Moreover, both isoflavones significantly inhibited the hepatic fatty acid β-oxidation [53,54]. Genistein was found to increase liver antioxidant enzyme activity, namely, superoxide dismutase, catalase, and glutathione peroxidase, in STZ-induced diabetic rodents leading to increased insulin sensitivity [36,55]. Similar results were obtained in type 2 diabetic animals (db/db mice), and in rat model of insulin resistance, which resulted in better glycemic control and improved plasma total cholesterol and triglyceride levels [40,56]. In the case of genistein, it was found that its ability to increase methylation in the regulatory regions of the genes encoding gluconeogenic enzymes might be responsible for the observed phenomena [24]. Administration of formononetin significantly increased hepatic glycogen level and improved insulin sensitivity in STZ-diabetic rats as well [45]. In turn, biochanin A could reverse the unfavorable changes in gluconeogenic enzyme activity in the liver of HFD C57BL/6J mice due to the increase in glycogen content [57].

Data on the influence of isoflavones on liver steatosis is less consistent. On the one hand, a high-isoflavone soy protein isolate diet protected rodents on HFD from NAFLD; on the other, supplementation of Zucker fatty rats with daidzein alone did not improve the liver steatosis scores [56,58,59,60]. Through the phosphorylation of AMPK, genistein ameliorated the accumulation of fat in fatty acid-induced Buffalo Rat liver (BRL) cells [61]. Another molecular mechanism involved in the protective effect of genistein against NAFLD is the inhibition of COX-1 activity and downstream TXA_2_ biosynthesis. By triggering this pathway, genistein efficiently normalized liver steatosis and glucose tolerance in diabetic mice [19,62]. It was also suggested that the inhibitory effect of genistein on liver steatosis is secondary to its ability to modify adipose tissue metabolism, namely, upregulation of fatty acid β-oxidation and downregulation of lipogenesis and pro-inflammatory cytokines expression [63]. Analogical effects were reported in the case of formononetin and biochanin A, which both prevented obesity-induced hepatic steatosis and insulin resistance in HFD and STZ-induced diabetic mice [62,64,65,66]. However, a synthetic formononetin analog (2-heptyl-formononetin) induced hepatic steatosis in C57BL/6J mice [67].

The efficacy of isoflavones in the prevention and treatment of NAFLD has also been assessed in humans. Xin et al. summarized eight randomized clinical trials (RCTs) including 424 individuals (394 female, 30 male), investigating the impact of genistein supplementation (54–250 mg/d, intervention time: 2–36 months) on NAFLD [68]. In five studies, treatment with genistein decreased fasting glucose and insulin levels; changes in liver function tests were not observed by any study. None of the studies included a hepatic histopathological evaluation to estimate the efficacy of the intervention. Therefore, the available evidence does not establish whether isoflavones are effective in NAFLD management in humans.

### 4.3. Muscle

Skeletal muscles (SM) account for approximately 80% of total body glucose disposal in humans, and insulin resistance in SM manifests itself as impaired glucose uptake following ingestion of a carbohydrate meal that results in postprandial hyperglycemia [30]. In T2D patients, the ability of insulin to stimulate glucose uptake is blunted compared to non-diabetic controls. This was confirmed by studies using an euglycemic insulin clamp and positron emission tomography (PET) scanning [69,70]. The pathogenesis of skeletal muscle insulin resistance in T2D patients is complex and involves intramyocellular lipids deposition, mitochondrial defects, and an ongoing low-grade inflammatory process [30].

Isoflavones have been demonstrated to improve SM insulin resistance in some animal models.

For instance, genistein promotes SM development by regulating the miR-222 level, increasing the expression of slow muscle fibers and promoting mitochondrial biogenesis in mice [27]. In addition, through AMPK activation and interaction with PPARδ genistein induced the expression of fatty acid oxidation genes in the SM of Zucker Diabetic Fatty (ZDF) rats and leptin receptor (ObRb) silenced C2C12 myotubes [71]. However, in another study, an eight week administration of genistein to ZDF rats did not influence SM metabolism and did not prevent the progression of the hyperinsulinemic normoglycemic state (pre-diabetes) toward full-blown T2D [72]. Similarly, a ten week consumption of isoflavone-containing soy protein (the equivalent of human intake of 75 mg/d) did not increase muscle PPARα or PPARγ expression and had little influence on insulin responses to the glucose challenge in cynomolgus monkeys [73].

Daidzein upregulated mitochondrial biogenesis in C2C12 myotubes, which was mediated by the activation of NrF and SIRT1 [74] and increased the expression of genes related to fatty acids oxidation such as pyruvate dehydrogenase kinase 4 (Pdk4) and acyl-coenzyme A dehydrogenase (Acadm). It also increased oxidative phosphorylation of ATP synthase F1 subunit beta (Atp5b) and cytochrome c (Cycs) through the activation of ERα. These actions resulted in decreased lipid accumulation and oxygen consumption in C2C12 myotubes [75]. In addition, daidzein promotes muscle glucose uptake inducing glucose transporter 4 (GLUT-4) translocation to the plasma membrane in L6 myoblasts in vitro. It improves AMPK phosphorylation in diabetic db/db mice muscles in vivo, which results in increased insulin sensitivity [76]. Administration of biochanin A to HFD-fed mice significantly upregulated the expression of insulin receptor substrate 1 (IRS 1), phosphoinositide 3-kinases (PI3K), and GLUT-4 in the skeletal muscle of HFD-induced diabetic mice when compared to HFD-diabetic mice that did not receive this isoflavone [57].

The influence of isoflavones on muscle function has also been evaluated in clinical trials. In a nine-month randomized clinical trial (RCT), genistein-based supplementation had no additive effect to resistance training on postmenopausal women’s body composition [77]. However, a recent study suggested the effect of soy isoflavones on muscle metabolism might depend on microbiome composition. A two-month administration of genistein increases the oxidative capacity of fatty acids in the skeletal muscle, thus improving insulin sensitivity in subjects with metabolic syndrome that was associated with a modification of the microbiome composition (particularly an increase in *Akkermansia muciniphila* content) [78]. This finding suggests that the impact of isoflavones on muscle insulin resistance depends on their metabolism by the gut microbiota.

### 4.4. Adipose Tissue

Estrogens are potent modulators of the proliferation and function of adipocytes. The effect of estrogens and phytoestrogens on adipocyte proliferation is not the subject of this review and is described elsewhere [79]. From the point of view of insulin resistance and diabetes, their influence on adipose tissue metabolism and secretory activity is of particular importance.

In vitro and in vivo studies confirmed that estradiol controls lipoprotein lipase (LPL) activity, an indicator of lipid storage. Its transdermal administration to humans decreases the expression of genes encoding critical lipogenic enzymes (stearoyl-CoA desaturase, fatty acid synthase, acetyl-coenzyme A carboxylase alpha, fatty acid desaturase 1) in subcutaneous adipose tissue deposits, which correlates with a decrease in plasma triglyceride (TG) levels [80]. Several isoflavones have been found to modulate adipose tissue lipolysis and lipogenesis. Genistein decreased basal and insulin-induced lipid synthesis in adipocytes isolated from the ovariectomized rats, 3T3-L1 preadipocytes, and rodents on HFD [63,81,82,83,84,85]. In turn, daidzein downregulated the expression of genes involved in lipids synthesis and upregulated genes involved in fatty acid β-oxidation in adipose tissue of C57BL/6J mice with diet-induced obesity [54,60,85]. The effectiveness of the daidzein-mediated inhibition of lipogenesis was also proved in clinical trials, where an eight week administration activated PPARγ and downregulated the expression of genes involved in lipids synthesis in adipose tissue; the effect was independent of the individual’s ability to produce equol, its active metabolite [86]. By suppressing cAMP phosphodiesterase, both genistein and daidzein were found to enhance lipolysis in rat adipocytes, decreasing the TG accumulation and preventing their hypertrophy [87,88]. Another molecular mechanism involved in genistein lipolytic properties is related to its ability to decrease miR-222 levels in adipose tissue [28]. Formononetin was also found to promote lipolysis and increase glycerol release in murine adipocytes in vitro [89]. Through the activation of PPARγ and subsequent increase in uncoupling protein 1 expression in adipocytes, formononetin and genistein enhance thermogenesis, which was shown to protect C57BL6/J from HFD-induced weight gain [90,91]. However, these actions of isoflavones do not always result in the increased adipose tissue insulin sensitivity [92].

Isoflavones modulate the secretory activity of adipocytes by decreasing lipid synthesis and enhancing lipolysis, since the accumulation of lipids leads to increased expression of genes encoding cytokines chemokines and adhesion molecules in these cells. This subsequently attracts infiltrating immune cells that further contributes to the synthesis of pro-inflammatory mediators and the development of a chronic inflammation related to excessive obesity, called “metaflammation” [93]. Several studies proved that isoflavones can modulate obesity-related inflammation in adipose tissue and its secretory profile through different molecular mechanisms [63,86,94,95,96,97,98].

Genistein interferes with the NF-κB pathway to downregulate both endogenous and TNFα-induced synthesis of IL-6 and IL-8 in 3T3-L1 preadipocytes and in C57BL/6 mice on HFD, that results in improved insulin sensitivity [63,94,95]. Similarly, biochanin A inhibits PPARγ and blocks MAPK phosphorylation to downregulate leptin, TNFα, and IL-6 expression in preadipocytes [96] and increases the adiponectin level in adipose tissue of STZ-induced diabetic rats [97]. In turn, by activating PPARα and PPARγ and inhibiting the c-Jun N-terminal kinase (JNK) pathway, daidzein was found to decrease the expression of MCP-1 and IL-6 and increase the expression of adiponectin in 3T3-L1 adipocytes and macrophages co-cultures [98]. The same changes in secretory activity of adipose tissue were observed in C57BL/6J mice with diet-induced obesity after daidzein treatment, which was accompanied by inhibited macrophage infiltration [63].

Notably, the anti-inflammatory properties of isoflavones were confirmed in clinical trials. For instance, daidzein-based supplementation in postmenopausal women led to the downregulation of *MAPK1* and *KRAS* genes expression in adipose tissue (both involved in the activation of the inflammatory response), with a simultaneous upregulation of genes encoding proteins with anti-inflammatory properties, such as IL-10 receptor antagonist (IL10RA) and NF-κB inhibitor α (NFKBIA) [86].

### 4.5. Kidneys

Even though the kidney filters approximately 160 g of glucose daily, a healthy subject’s urine does not contain glucose at all. That is because 90% percent of the filtered glucose is reabsorbed by the high capacity sodium-glucose cotransporter-2 (SGLT2) in the convoluted segment of the proximal tubule, while the remaining 10% is reabsorbed by the SGLT1 in the straight segment of the descending proximal tubule. Cultured human proximal renal tubular cells from T2D patients demonstrate markedly increased levels of SGLT2, which means the kidney of a diabetic patient increases glucose reabsorption instead of excreting excess glucose to correct the hyperglycemia [3]. Apart from this critical disturbance of glucose homeostasis, in the progression of diabetes a chronic loss of kidney function occurs. Diabetic nephropathy is one of the leading causes of chronic kidney disease and end-stage renal disease globally. Higher glucose reabsorption capacity in proximal renal tubules is associated with higher urinary albumin excretion and low glomerular filtration rate in patients with diabetes. Therefore, apart from the direct inhibition of SGLTs, the prevention of chronic kidney disease contributes to the reduction of renal glucose reabsorption in a diabetic patient [99].

Animal studies suggest that isoflavones can favorably modify the course of diabetic nephropathy. Treatment with genistein was found to inhibit oxidative stress through activation of Nrf2 and attenuate renal fibrosis and inflammation by suppression of the *transforming growth factor*-beta—TGF-β and NF-κB in STZ-induced diabetic rodents, alloxan-induced diabetic mice, and the experimental model of the nephrotic syndrome [100,101,102,103]. Similarly, administration of daidzein to STZ-induced diabetic rats led to a significant reduction of creatinine and blood urea nitrogen levels and an increase of antioxidant enzymes (glutathione, catalase, and superoxide dismutase) expression and amelioration of histological changes in kidneys [104,105]. A 16-week formononetin treatment led to an improvement of glycemic parameters and significantly enhanced creatinine clearance in the same experimental model that was associated with increased SIRT1 expression in the kidney [106]. Moreover, isoflavones supplementation lowered urinary albumin excretion and decreased the urine albumin-to-creatinine ratio, which delayed the progression of diabetic nephropathy in diabetic *db/*db** mice. However, this intervention had little influence on plasma glucose concentrations [107,108,109].

Human studies on the influence of soy supplementation and renal function reported inconsistent findings. Several short-term (about eight week long) clinical trials in T2D patients with diabetes indicated that substituting soy protein for animal protein decreases hyperfiltration and albuminuria, therefore slowing renal function deterioration [110,111]. However, in a six month RCT, neither whole soy nor daidzein supplementation influenced renal function [112]. Further studies are required to determine the role of isoflavones in the progression of diabetic nephropathy.

### 4.6. Gastrointestinal Tract

Glucose-dependent insulinotropic peptide (GIP) and glucagon-like peptide-1 (GLP-1) are incretins—hormones secreted from enteroendocrine gut cells into the blood in response to food ingestion, modulating the insulin secretory response depending on the nutrient content of the meal. The response of β-cells to incretins is called the incretin effect and accounts for approximately 50% of the total insulin secretion after oral glucose consumption. In T2D, pancreatic islet response to incretins is impaired; for instance, GIP no longer modulates glucose-dependent insulin secretion, even at supraphysiological doses. Given that incretins are also trophic factors for β-cells, unresponsiveness to incretins may contribute to islet degeneration [113]. GLP-1 analogs and inhibitors of dipeptidyl peptidase 4 (DPP4)—an enzyme responsible for incretin degradation—were registered for T2D treatment over a decade ago. In recent years, GLP-1 analogs in particular have become important in diabetes treatment due to their cardioprotective properties. Apart from the stimulation of insulin release, the anti-diabetic effects of these drugs include suppression of glucagon secretion, delay in gastric emptying, and inhibition of small bowel motility. There is a growing interest in natural substances that could increase the secretion or potentiate the incretin effect.

Both genistein and daidzein stimulated GLP-1 secretion in enteroendocrine colorectal cancer cells in vitro [114]. This phenomenon was confirmed in vivo, since genistein alone or in combination with metformin downregulated the inflammatory response and was found to enhance GLP-1 secretion in the murine intestine [115]. Moreover, administration of genistein to alloxan-induced diabetic mice protected intestinal L-cells (responsible for GLP-1 secretion) from alloxan-caused damage [115]. Interestingly, treatment with a plant extract containing daidzein increased endogenous GLP-1 and GIP levels in STZ diabetic rats by inhibition of DPP4, triggering the mechanism used in the pharmacological management of diabetes [116].

The hypoglycaemic properties of isoflavones in the gastrointestinal are not limited to the modulation of incretin secretion. For instance, genistein was also found to inhibit yeast α-glucosidases—enzymes responsible for carbohydrate digestion [117]. Nevertheless, yeast α-glucosidase differs from the mammalian form, and this finding should be verified in other experimental models.

Some of the beneficial health effects attributed to isoflavones could result from their ability to modulate gut microbiota populations. When tested in vitro, genistein reduced the abundance of *Bacteroides fragilis*, *Lactococcus lactis,* and *Slackia equolifaciens*, while both genistein and equol increased the amount of *Lactobacillus rhamnosus* and *Faecalibacterium prausnitzii* [118]. The results of studies on the influence of isoflavones supplementation on the composition of the microbiome vary depending on the studied population and the type of intervention. For instance, a five day consumption of soy products (the combination of glycitein, genistein, and daidzein) in equol-producers decreased the abundance of *Clostridium* clusters and increased the number of bifidobacteria [119]. In turn, six months of isoflavone supplementation resulted in an increased abundance of *Clostridium* and decreased bifidobacteria and enterobacteria populations in equol producers, while the opposite trend was observed in non-producers [120]. HFD leads to changes in microbiota composition and increased intestinal permeability. Subsequently, intestinal bacteria translocate into the systemic compartment, which is associated with elevated circulating lipopolysaccharide (LPS) and inflammatory markers levels, and increased risk of diabetes both in experimental animals and humans [121,122]. A six-month administration of genistein to HFD mice modified their gut microbiota, and this was associated with lower circulating LPS levels, reduced expression of pro-inflammatory cytokines in the liver, and improved insulin sensitivity [55,123]. Similar results were obtained in NOD mice [124]. In obese subjects with metabolic syndrome and insulin resistance, a two-month treatment with genistein (50 mg/day) significantly improved insulin sensitivity, and the effect was associated with a modification of the gut microbiota (namely an increase in *Akkermansia muciniphila*) and an increase in oxidative capacity of fatty acids in the skeletal muscle [78].

### 4.7. Brain

Insulin provides satiety signals to the brain, but in obese individuals with insulin resistance the appetite regulation provided by insulin is impaired [3]. Cerebral insulin resistance leading to increased hepatic glucose production and reduced muscle glucose uptake was confirmed in animal models [125].

There is some evidence from animal studies that isoflavones can improve whole-body energy homeostasis and decrease the risk of T2D by regulating gene expression in the hypothalamus. In C57BL/6 mice, the addition of genistein to HFD resulted in improved glucose tolerance and insulin sensitivity compared to the control animals on HFD only after eight weeks. These changes were associated with altered expression of *Ucn3, Depp*, and *Stc1* in the hypothalamus and significantly correlated with adipose tissue browning and insulin sensitivity [126]. Daidzein showed a similar, though slightly less pronounced, effect on appetite and glucose tolerance in HFD C57BL/6 mice and in ovariectomized ObRb deficient rats [127,128]. A molecular mechanism responsible for the anorectic properties of isoflavones involves the reduction of hypothalamic oxidative stress and neuroinflammation [129].

In a clinical trial, the eight-week administration of 50 mg of soy isoflavones daily (the upper range of a traditional Asian diet) significantly increased peptide YY (a satiety hormone) in healthy postmenopausal women independent of the ability to produce equol. However, this effect did not translate into reduced energy intake and weight gain [130]. Additionally, in rats daidzein stimulated cholecystokinin (CCK) secretion in the small intestine, suggesting that CCK is involved in the hypothalamic regulation of its anorectic effect [131,132]. However, it should be mentioned that not all isoflavones stimulate satiety. For instance, biochanin A was found to inhibit the metabolism of the endogenous cannabinoid receptor ligand anandamide in vitro, which suggests its orexigenic properties [133].

The mechanisms by which isoflavones can favorably modulate the function of organs crucial for T2D pathogenesis are summarized in Table 1.

## 5. Isoflavones in Management of Type 2 Diabetes

The effectiveness of isoflavones for the prevention and treatment of T2D and related complications has been assessed in several epidemiological and clinical studies. Epidemiological studies tried to find correlations between dietary isoflavones intake and the occurrence and progression of T2D. Clinical trials investigated how isoflavones supplementation may modulate the progression of diabetes and related complications.

### 5.1. Epidemiological Studies

The concept of the influence of isoflavones on the development of T2D comes from the observation that the incidence of the disease is significantly lower in Asian populations compared to European and North American populations. Considering the role of lifestyle in the development of T2D, it was noted that a factor that differs between the diet of Asian populations from those of Western countries is the higher content of soy products [134]. Subsequently, several studies investigated how soybean products consumption might influence the incidence of T2D.

In three prospective cohort studies (Nurses’ Health Study, Nurses’ Health Study II, and Health Professionals Follow-Up Study) involving 163,457 individuals, an inverse association between dietary isoflavones intake and T2D risk was observed [135]. Higher plasma genistein levels were also associated with a lower T2D risk in the case-control Korean Genome and Epidemiology Study [136]. Similarly, in a study including 299 pregnant women (participants of the NHANES 2001–2010 study), urinary concentrations of total isoflavone metabolites were inversely associated with fasting plasma glucose (FPG), insulin, and homeostatic model assessment of insulin resistance (HOMA-IR). In this cross-sectional study, a significant inverse association between urinary equol level and FPG was observed [137]. In principle, the Singapore Chinese Health study, which included 564 diabetic patients and 564 healthy controls, had the opposite result, since no significant associations were found between the total urine isoflavones and risk of T2D. However, when the associations between subclasses of urine metabolites of isoflavones were analyzed, the second quartile of daidzein and third quartile of genistein were significantly associated with a lower risk of T2D compared with the corresponding first quartile [138]. It is important to note that the abovementioned studies were conducted in regions with different dietary habits. Consumption of soy products is generally lower in the Western diet, leading to the modest effect of isoflavones on metabolic markers. In a typical Western diet, lignans, not isoflavones, are the primary form of phytoestrogens. Moreover, the source of dietary isoflavones is also essential, since in the Shanghai Women’s Health Study (involving over 64,000 participants), only soy milk consumption was associated with a lower risk of T2D. There was no significant association between consumption of other soy products or total soy protein intake and the risk of diabetes [139].

The results of epidemiological studies assessing the effect of isoflavones on T2D risk may be influenced by the sex and hormonal status of study participants. For instance, in the abovementioned Shanghai Women’s Health study, no overall statistically significant association between risk of glycosuria and soy intake was found. However, when menopausal status was taken into account, soy protein consumption (8 g/d) was inversely associated with glycosuria in normal-weight postmenopausal study participants, after adjusting for age, education, body mass index, physical activity, hormone replacement therapy, total caloric intake, and other dietary factors [140]. By contrast, in the Nutrition and Health of Aging Population in China project, similar soy protein intake was associated with increased risk of hyperglycemia in men [141]. Therefore, it seems that isoflavone effects on glucose metabolism and insulin sensitivity are sex-specific, resulting from the estrogen-like activity of the compounds. A similar relationship can be observed in the case of testosterone—high concentrations are associated with a lower risk of T2D in men but with a higher risk in women [142].

It should be noted that there are important differences between epidemiological studies in parameters and diagnostic tests used for the assessment of glycemic status of study participants, which may influence results. While previous studies’ conclusions were based on FPG, glycosuria, and glycated hemoglobin (HbA1c) levels, in a study performed in Japan, which found that high consumption of soybean products and vegetables was positively associated with impaired glucose tolerance, T2D diagnosis was based on the 75 g oral glucose tolerance test [143]. Nevertheless, six meta-analyses conducted between 2017 and 2020 confirmed an inverse association between the dietary isoflavones intake and T2D risk [144,145,146,147,148,149].

### 5.2. Interventional Studies

The encouraging results of animal studies prompted researchers to evaluate whether isoflavones could prevent T2D in humans and affect the progression of diabetes. Several clinical trials were conducted and they differed in terms of sample size, age, sex, ethnicity, and hormonal status of participants, duration and type of intervention, and the endpoints assessed. As a result, the findings were often contradictory.

Over 30 years ago, Tsai et al. observed that in seven obese subjects with T2D, the daily consumption of 10 g of soy polysaccharide significantly reduced postprandial serum glucose and triacylglycerol concentrations. This effect appeared to result from a lower glucagon secretion, since the supplementation had no significant effect on serum insulin concentrations [150]. Several further studies supported this finding [151,152,153,154,155]. For instance, in postmenopausal women with T2D, one-year of isoflavone intake (100 mg of aglycones) improved insulin sensitivity and the lipid profile [151]. Similarly, 132 mg of isoflavones administered daily for 12 weeks significantly reduced fasting insulin, HOMA-IR, and HbA1c in postmenopausal women with diet-controlled T2D [152]. However, in another study, a three month supplementation with the same dose of isoflavones had no impact on the abovementioned parameters [156]. Several other RCTs did not confirm the favorable effects of isoflavones on glycemic control and insulin sensitivity in postmenopausal women [157,158,159,160,161]. Since these discrepancies do not depend on equol-production status [159], the study design seems to have had a significant impact on the results obtained. Consequently, the influence of isoflavones on the prevalence and course of T2D has been analyzed in several meta-analyses.

In a meta-analysis covering 24 RCTs and 1518 subjects (of different sexes, ages, and glycemic status), no effect of soy intake on FPG and insulin concentrations was observed. However, subgroup analysis revealed a favorable change in FPG level in studies that used whole soy foods or a soy diet [162]. A meta-analysis of 12 trials including a more homogenous group (perimenopausal and postmenopausal non-Asian women) found that the intake of soy isoflavones was associated with a significant glycemia and fasting insulin reduction, mostly when the intervention time was longer than six months [163]. In a more recent meta-analysis, which evaluated the impact of phytoestrogens on body composition (23 RCTs postmenopausal women of different ethnicity, 1130 in the intervention group, and 750 controls), isoflavone intake was associated with lower FPG levels independent of weight. The effect was more pronounced in the low-dose (<100 mg/d) and longer (≥6 months) treatment, but only in non-obese study participants. By contrast, isoflavones supplementation in obese women with T2D was associated with an increase in body weight and, in the case of daidzein, with unfavorable body composition changes [164]. A year ago, the results of another meta-analysis of 16 RCTs (six parallel and ten cross-overs) evaluating the effects of soy consumption on glucose metabolism in T2D patients were published [165]. This meta-analysis found that in the whole studied group (471 participants), soy consumption did not significantly improve FPG, fasting insulin, or HbA1c levels in patients with T2D compared with a placebo. However, when only parallel studies were analyzed, soy consumption positively affected FPG and fasting insulin concentrations. The authors indicated the potential difficulties associated with analyzing data concerning the influence of isoflavones on the prevalence and course of T2D. Apart from differences in the type of compound administered and the intervention period, the analyzed studies were heterogeneous in terms of patient characteristics (age, disease duration), and only a few accounted for equol-production status. Therefore, more high-quality studies are required to reach reliable conclusions.

Described above epidemiological studies, clinical trials, and meta-analyses regarding the influence of isoflavones on T2D progression are summarized in Table 2.

## 6. Final Remarks and Conclusions

With the world struggling in the face of an obesity pandemic, the number of people suffering from T2D is growing every year, since diabetes is a common metabolic complication of obesity. Given the social and financial costs associated with the management of diabetes itself and diabetes-associated disability, every effort should be made to improve the methods of its prevention and treatment. Therefore, in addition to the development of novel pharmacological strategies, there is an ongoing search for natural dietary compounds that could be used for this purpose.

In the case of isoflavones, the results of preclinical studies are promising, because they showed a beneficial effect on the function of critical organs involved in T2D pathogenesis. In several of the in vitro studies and animal models described here, isoflavones prevented β-cell destruction and increased insulin secretion while reducing glucagon production; reduced lipolysis and inflammation in the liver, which translated into increased insulin sensitivity, improved glucose uptake in the muscle, and favorably modified metabolism and secretory activity of adipose tissue. Besides the impacts on organs traditionally associated with the development of hyperglycemia, isoflavones have been shown to reduce the progression of chronic kidney disease and affect glucose reabsorption, potentiate incretins action, modify the composition of the intestinal microflora, and increase hypothalamic appetite control. However, the observed phenomena (summarized in Table 1) refer to different experimental models and conditions, and the concentrations of isoflavones used in preclinical studies are much higher than those that occur naturally in living organisms. Interpretation of preclinical studies results is also hampered by a lack of systematic reviews that would identify, appraise, and synthesize all relevant data on this topic.

While the meta-analyses of epidemiological studies suggest an inverse association between isoflavones dietary intake and T2D risk, the results of clinical trials published over the last 30 years evaluating the effects of isoflavones on T2D risk in humans are contradictory (summarized in Table 2). These discrepancies can be partly explained by the fact that the currently published studies are heterogeneous in terms of the study group size and characteristics, the type of intervention used, and the assessed endpoints. Therefore, further well-designed preclinical studies and randomized clinical trials followed by systematic reviews are needed to definitively assess the value of isoflavones for the prevention and treatment of diabetes. Considering the study participants’ hormonal status and the composition of their intestinal microflora might be crucial to obtain reliable results.

## Figures and Tables

**Figure 1 ijms-22-00218-f001:**
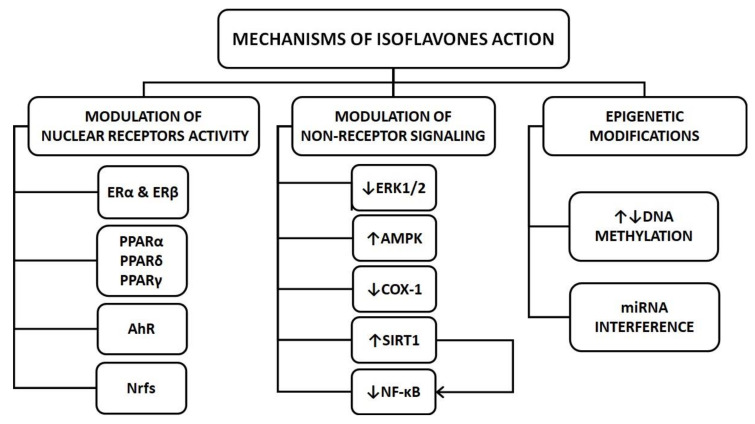
Selected mechanisms of isoflavones action relevant to the pathogenesis of type 2 diabetes. AMPK—adenosine monophosphate-activated protein kinase, AhR—aryl hydrocarbon receptor, COX-1—cyclooxygenase 1, ER—estrogen receptor, ERK1/2—extracellular signal-regulated kinases 1 and 2, miRNA—microRNA, NF-κB—nuclear factor κB, Nrfs—nuclear respiratory factors, PPAR—peroxisome proliferator-activated receptors, SIRT1—sirtuin 1, ↑—increase, ↓—decrease.

**Table 1 ijms-22-00218-t001:** Mechanisms by which isoflavones can favorably modulate the function of organs crucial for the pathogenesis of type 2 diabetes.

	Genistein	Daidzein	Formononetin	Biochanin A	Molecular Mechanism	Experimental Model	Reference
**β-cells**	↑insulin secretion			↑insulin secretion	↓NF-κB ↓ERK-1/2	RIN cells	[20] [33]
	↑proliferation					STZ rats	[38]
	↓DNA fragmentation				↑ERβ	human β-cells	[31]
	↑↓apoptosis *				↑↓ERK1/2	rodent β-cells	[32]
	↑proliferation	↓apoptosis		↑proliferation	↑AMPK	INS-1 cells	[33,42,47]
		↓apoptosis			↓PKA ↑PP2A	INS-1 cells ZDF rats	[43]
	↑insulin secretion ↓apoptosis	↑insulin secretion			↑AMPK ↑calcineurin	INS-1 cells mouse pancreatic islets STZ/HFD mice db/db mice	[35,36] [34] [37,38,39] [40]
	↓inflammation		↑insulin secretion	↑insulin secretion*	↑SIRT1	alloxan-induced diabetic mice ovariectomized rats STZ rats	[44] [41] [47,48]
			↓apoptosis		↓NF-κB	INS-1 cells STZ rats	[46] [45]
**α-cells**		↓glucagon secretion			?	geese broilers db/db mice	[49] [50] [40]
			=glucagon secretion			alloxan-induced diabetic mice	[44]
**liver**	↓gluconeogenesis ↑fatty acid oxidation	↓gluconeogenesis ↑fatty acid oxidation			↑methylation	NOD mice HFD diabetic mice monkeys db/db mice insulin resistant rats	[53] [54] [24] [40] [56]
			↓glycogenolysis			STZ rats	[45]
	↑antioxidant activity	↑antioxidant activity				STZ rats and mice	[37,55]
				↓gluconeogenesis		HFD diabetic mice	[57]
	=steatosis	=steatosis				ZDF rats	[59]
			↓steatosis	↓steatosis	↑PPARα	HFD diabetic mice	[66]
	↓steatosis	↓steatosis	↓steatosis	↓steatosis	↑AMPK	HFD diabetic mice	[60,63,64,66]
			↓steatosis		↓COX-1	HFD diabetic rodents db/db mice	[19,56,58,65] [62]
	↓steatosis					Buffalo Rat liver	[61]
**muscle**	↑proliferation				↓miR-222	C2C12 myotubes	[27]
	↑mitochondrial biogenesis	↑mitochondrial biogenesis ↑fatty acid oxidation			↑SIRT1 ↑NrF ↑ERα		[74] [75]
	↑fatty acid oxidation	↑insulin sensitivity			↑AMPK ↑ PPARδ ↑GLUT4	C2C12 myotubes ZDF rats db/db mice	[71] [76]
	= insulin sensitivity	= insulin sensitivity			=PPARγ =PPARα	ZDF rats cynomolgus monkeys	[72] [73]
		↑insulin sensitivity		↑insulin sensitivity	↑IRS1 ↑GLUT4	HFD diabetic mice	[57]
	↑insulin sensitivity					MetS patients	[77,78]
**adipose tissue**	↓lipid synthesis		↑lipolysis ↑thermogenesis		↑PPARγ	3T3-L1 preadipocytes ovariectomized rats C57BL/6 obese mice obese Wistar rats	[82,95] [81,84] [63,90,91] [92]
	↓lipid synthesis	↓lipid synthesis			↑AMPK	3T3-L1 preadipocytes HFD rodents human AT	[85] [54] [86]
	↑lipolysis	↑lipolysis			↓cAMP PDE	rat adipocytes C57BL/6 obese mice	[87,88] [63]
	↑lipolysis				↓miR-222	3T3-L1 preadipocytes HFD mice	[28]
	↓inflammation				↓NF-κB	3T3-L1 preadipocytes	[94]
		↓inflammation			↑PPARα ↑PPARγ	C57BL/6 obese mice 3T3-L1 preadipocytes human AT	[63] [98] [86]
				↓inflammation ↑adiponectin	↓PPARγ	rat adipocytes STZ rats	[96] [97]
**kidney**	↓oxidative stress ↓fibrosis ↓inflammation	↓oxidative stress ↓fibrosis			↑NrF ↓TGF‑β ↓NF-κB	STZ rodents	[100,101,102,104,105]
	↓albuminuria ↑GFR	↓albuminuria ↑GFR	↓albuminuria ↑GFR		↓smad3	db/db mice	[107,108,109]
	↓albuminuria ↑GFR	↓albuminuria ↑GFR				T2D patients	[110,111]
		=albuminuria =GFR				T2D patients	[112]
			↑GFR		↑SIRT1	STZ rats	[106]
**gastrointestinal tract**	↑GLP-1	↑GLP-1			↓NF-κB	enteroendocrine cells alloxan-induced diabetic rats STZ rats	[114,115] [115] [116]
	↓inflammation	↓inflammation			modulation of gut microbiota	HFD mice NOD mice	[121,122,123] [124]
**brain**	↑insulin sensitivity ↓oxidative stress ↓inflammation	↑insulin sensitivity ↓oxidative stress ↓inflammation			modulation of hypothalamic gene expression	HFD mice ObRb(-)rats	[126,127] [128]
	↓appetite	↓appetite			peptide YY	healthy women	[130]
				↑appetite		Wistar & Sprague-Dawley rats	[132,133]

AMPK—adenosine monophosphate-activated protein kinase; AT—adipose tissue; cAMP PDE—cAMP phosphodiesterase; COX-1—cyclooxygenase 1; db/db mice—genetically obese diabetic; ER—estrogen receptor; ERK-1/2—extracellular signal-regulated kinases 1 and 2; GFR—glomerular filtration rate; GLUT-4—glucose transporter 4; HFD—high-fat diet; INS-1 cells—rat insulinoma cells; MetS—metabolic syndrome; NAFLD—non-alcoholic fatty liver disease; NF-κB—nuclear factor κB; NrF—nuclear respiratory factor; NOD mice—non-obese diabetic mice; ObRb—leptin receptor; PKA—protein kinase A; PP2A—protein phosphatase; PPAR—peroxisome proliferator-activated receptor; RIN cells—rat islet cells; SIRT1—sirtuin 1; STZ rats/mice—streptozotocin-induced diabetic rats/mice; STZ/HFD mice—STZ treated mice on a high-fat diet; T2D—type 2 diabetes; ZDF rats—Zucker Diabetic Fatty rats. ↑—increase; ↓—decrease; ?—unknown mechanism; = no change; * depending on the concentration.

**Table 2 ijms-22-00218-t002:** Epidemiological studies, clinical trials and meta-analyses regarding the influence of isoflavones on type 2 diabetes (T2D) course and progression.

Study Design	Studied Group	Measurements	Intervention	Outcomes	Reference
**epidemiological studies**					
NHS NHSII HPFS prospective	163,457 individuals (142176 F) 9181 T2D cases (8439 F)	soy and isoflavones intake assessed with a validated FFQ	not applicable	inverse association between isoflavones intake and T2D risk	[135]
Korean Genome and Epidemiology Study case-control	698 healthy controls (317 F) 693 T2D patients (316 F)	plasma levels of isoflavones, genistein, daidzein, equol, glyctein	not applicable	inverse association between plasma genistein level and T2D risk	[136]
NHANES 2001–2010 cross-sectional	299 healthy pregnant women	urinary concentrations of total isoflavones metabolites	not applicable	inverse association between urinary concentrations of total isoflavone metabolites with FPG, insulin, HOMA-IR	[137]
Singapore Chinese Health Study cross-sectional	564 healthy controls (329 F, 192 postmenopausal) 564 T2D patients (329 F, 199 postmenopausal)	urinary concentrations of total isoflavones metabolites	not applicable	inverse association between daidzein and genistein intake and T2D risk	[138]
Shanghai Women’s Health Study prospective	64 227 pre- and postmenopausal women 1608 T2D cases	legumes intake assessed with a validated FFQ (follow-up survey after 2–3 years)	not applicable	inverse association between soybeans intake and T2D incidence	[139]
Shanghai Women’s Health Study	39 385 postmenopausal women 323 T2D cases	soy products intake assessed with a validated FFQ	not applicable	inverse association between soy products intake and risk of glycosuria	[140]
Nutrition and Health of Aging Population in China project cross-sectional	2811 individuals (1638 F)	soy protein intake assessed with a validated FFQ	not applicable	positive association between soy protein intake with hyperglycemia in men	[141]
**meta-analyses of epidemiological studies**					
NHS NHSII Singapore Chinese Health Study Korean Genome and Epidemiology Study	144,695 individuals 9696 T2D cases	soy and isoflavones intake assessed with a validated FFQ; urine/plasma levels of isoflavones, genistein, daidzein, equol, glyctein	not applicable	inverse association between isoflavones, genistein, and daidzein and T2D incidence	[144]
6 prospective cohort studies	299,667 individuals 24,469 T2D cases	isoflavones intake assessed with a validated FFQ	not applicable	inverse association between isoflavones intake and T2D incidence	[145]
6 prospective cohort studies 2 cross-sectional studies	1,901,230 individuals 7589 T2D cases	soy products and isoflavones intake assessed with a validated FFQ	not applicable	inverse association between isoflavones intake and T2D incidence	[146]
8 prospective cohort studies	312,015 individuals 19,953 T2D cases	total flavonoids, anthocyanidins, flavan-3-ols, flavolons and isoflavones intake assessed with a validated FFQ	not applicable	inverse and dose-dependent association between total flavonoids, anthocyanidins, flavan-3-ols, flavolons and isoflavones intake and T2D incidence	[147]
9 prospective cohort studies	172,058 individuals 16,910 T2D cases	flavanols, flavonols, flavan-3-ols and isoflavones intake assessed with a validated FFQ	not applicable	inverse and dose-dependent association between flavanols, flavonols, flavan-3-ols and isoflavones intake and T2D incidence	[148]
15 prospective cohort studies	565,810 individuals 32,093 T2D cases	legumes, total soy, soy milk, tofu, soy proteins and soy isoflavones intake assessed with a validated FFQ	not applicable	inverse association between tofu, soy proteins and soy isoflavones intake and T2D incidence	[149]
**interventional studies**					
RCT 2 arms cross-over (3-week washout period separating the placebo and active phases 12 weeks each)	32 postmenopausal women with T2D	FPG fasting insulin HOMA-IR HbA1c TC HDL-C LDL-C	132 mg isoflavones/day	↓fasting insulin ↓HOMA-IR ↓HbA1c ↓TC ↓LDL-C ↓TC/HDL-C ↓fT4	[152]
RCT 2 arms parallel (1 year)	93 postmenopausal women with T2D	FPG fasting insulin HOMA-IR TC HDL-C LDL-C	850 mg flavan-3-ols/day 100 mg isoflavones/day	↓HOMA-IR ↓fasting insulin ↓HDL-C ↓LDL-C	[151]
RCT 2 arms parallel (1 year)	120 postmenopausal women with MetS	FPG fasting insulin HOMA-IR TC HDL-C LDL-C TG adiponectin	54 mg genistein/day	↓FPG ↓fasting insulin ↓HOMA-IR ↓TC ↑HDL-C ↓LDL-C =TG ↑adiponectin	[153]
RCT 2 arms parallel (3 months)	200 men with T2D and subclinical hypogonadism	testosterone TSH fT4 HbA1c HOMA-IR TG CRP	66 mg isoflavones/day	=testosterone ↑TSH ↓fT4 ↓HbA1c ↓HOMA-IR ↓TG ↓CRP	[154]
RCT 2 arms parallel (12 weeks)	54 postmenopausal women with T2D	FPG fasting insulin HOMA-IR HDL-C TG	108 mg genistein/day	↓FPG ↓fasting insulin ↓HOMA-IR ↑HDL-C ↓TG	[155]
RCT 2 arms cross-over (4-week washout period separating the placebo and active phases 12 weeks each)	26 postmenopausal women with T2D	FPG fasting insulin HOMA-IR HbA1c TC TG HDL-C LDL-C CRP	132 mg isoflavones/day	=FPG =fasting insulin =HOMA-IR =HbA1c =TC =TG =HDL-C =LDL-C =CRP	[156]
RCT 2 arms cross-over (3-week washout period separating the placebo and active phases 6 weeks each)	14 male, 6 female T2D patients	FPG fasting insulin postprandial glucose postprandial insulin TC HDL-C LDL-C TG apoB100 homocysteine	165 mg isoflavones/day	=FPG =fasting insulin =postprandial glucose =postprandial insulin ↓TC ↓LDL-C ↓LDL-C/HDL-C ↓apoB100 ↓TG ↓homocysteine	[157]
RCT 3 arms parallel (6 months)	180 postmenopausal women with prediabetes or early untreated T2D	FPG 2h post-load glucose insulin HOMA-IR HbA1c	15 g soy protein + 100 mg isoflavones, 15 g milk protein +100 mg isoflavones, 15 g milk protein	=FPG =2h post-load glucose =fasting insulin =HOMA-IR =HbA1c	[158]
RCT 2 arms cross-over (3-week washout period separating the placebo and active phases 6 weeks each)	117 healthy post-menopausal women	FPG fasting insulin HOMA-IR HbA1c TC TG HDL-C LDL-C CRP	50 mg isoflavones/day	=FPG =fasting insulin =HOMA-IR =HbA1c =TC =TG =HDL-C =LDL-C ↓CRP	[159]
RCT 2 arms parallel (12 weeks)	75 healthy postmenopausal women	FPG fasting insulin HOMA-IR adiponectin resistin leptin	160 mg of total isoflavones (64 mg genistein, 63 mg daidzein, and 34 mg glycitein)	=FPG =fasting insulin =HOMA-IR ↑adiponectin =resistin =leptin	[160]
RCT 2 arms cross-over (4-week washout period separating the placebo and active phases 57 days each)	29 individuals with T2D (men and postmenopausal women)	FPG postprandial glucose fasting insulin postprandial insulin HOMA-IR	88 mg of total isoflavones (65% genistein, 31% daidzein and 4% glycitein)	=FPG =postprandial glucose =fasting insulin =postprandial insulin =HOMA-IR	[161]
**Meta-analyses of interventional studies**					
24 RCTs: 18 on women, (14 on postmenopausal) 6 on T2D patients 15 parallel design 9 cross-over	1518 subjects of different nationality, sexes, ages, and glycemic status	FPG fasting insulin HOMA-IR	soy-protein: 0–40 g/d, isoflavone content: 36–132 mg/d, time: 4–52 weeks	whole group: =FPG =fasting insulin subgroup analysis: whole soy foods ↓FPG	[162]
12 RCTs 4 on normal-weight women 8 on obese women	non-Asian postmenopausal women 601 intervention arm for FPG 581 placebo arm for FPG 582 intervention arm for fasting insulin 561 placebo arm for fasting insulin	FPG fasting insulin	isoflavone content: 40 to 160 mg time: 8–52 weeks	↓FPG ↓fasting insulin	[163]
23 RCTs: 19 on healthy women 4 on T2D women	1880 postmenopausal women of different nationality: 1130 intervention arm 750 placebo arm	BMI WHR	soy-protein:25–40 g/d, isoflavone content: 60–270 mg/d, time: 8–48 weeks	↑BMI in women with pre-existing: - prediabetes, - T2D, - prehypertension, - hyperlipidemia	[164]
16 RCTs: 6 parallel 10 cross-over	471 T2D patients (315 F) 43% of women postmenopausal	FPG fasting insulin HbA1c	soy-protein: 0.8–50 g/d, isoflavone content: 32–165 mg/d, time: 4 weeks–4 years	=FPG in 14 RCTs =insulin levels in 11 RCTs =HbA1c in 13 RCTs	[165]

apoB100—apolipoprotein B100, BMI—body mass index, CRP—C reactive, protein, FFQ—food-frequency questionnaire, FPG—fasting plasma glucose, F—female, fT4—free thyroxine, HBA1c—glycated hemoglobin, HDL-C—high-density lipoprotein cholesterol, HOMA-IR—homeostatic model assessment of insulin resistance, HPFS—Health Professionals Follow-Up Study, LDL-C—low-density cholesterol, MetS—metabolic syndrome, NHS—Nurses’ Health Study, NHSII—Nurses’ Health Study II, RCT—randomized control trials, R.R.—relative risk, T.C.—total cholesterol, T.G.—triglycerides, WHR—waist-hip ratio. ↑—increase; ↓—decrease; = no change.

## Data Availability

The data presented in this study are openly available in the PubMed database (www.pubmed.ncbi.nlm.nih.gov).

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
