# Peer review of "The Role of Isoflavones in Type 2 Diabetes Prevention and Treatment—A Narrative Review"

_ijms, 2020, doi:10.3390/ijms22010218_

Round 1

Reviewer 1 Report

The manuscript entitled “Isoflavones – a dietary option for type 2 diabetes prevention and treatment?” presents interesting issue, but some areas must be corrected.

Title:

Authors should formulate more “scientific” title - formulated while using a proper scientific language, as their current title is rather formulated as for the column of the newspaper. The proper title should be rather informative than catchy. The question mark is here hard to understand.

Language:

The manuscript is not prepared while using an adequate scientific language, so it should be corrected. The style should be polished, as some sentences are so hard to follow that they are even hard to understand – e.g. “With over 1.5 million deaths a year and being the leading cause of blindness, chronic renal failure, cardiovascular diseases, and lower limb amputation, T2D is a serious health and social problem and an economic one”.

It seems that Authors is not a native English speaker, so the whole manuscript should be corrected by a native English speaker, preferably by the professional agency.

Moreover, the manuscript is shabbily prepared with various fonts, sub-chapters not numbered consequently, etc.

Major:

There are really important problems associated with the prepared manuscript, including:

  • The serious flaw of the presented manuscript is associated with the fact, that it presents a highly subjective review, not a systematic review. While the systematic review has a key role for broadening knowledge, the other reviews don’t have such role.
  • Taking into account, that the Materials and methods section is not presented (it should be added), without any specific information, it is hard to understand which studies were included into review and why. Authors did not present any key words, which were used during literature search, inclusion and exclusion criteria of references, information about the procedure of literature search conducted by them, number of chosen references, as well as information if some of them were excluded from the review and on the basis of which criteria. As a number of recent publications that are related to the issue were not included, it is a serious problem.
  • Authors do not present the current and comprehensive knowledge associated with the issue. It is associated with the fact that they did not include some important issues, while other were included even if they are not so crucial
  • Including the reviews and meta-analysis into own review is also a highly controversial procedure – in many aspects, Authors just repeated the conclusions of other authors, without own analysis or conclusions.

Reviewer 2 Report

This is a review article regarding the effects of isoflavones on glucose metabolism. The manuscript was organized and written well. However, the data are mostly based on preclinical studies and there are few clinical evidence. Therefore, it is recommended that the conclusion should be less emphasized.

Graphical abstract seems too optimistic, too. This should be more realistic.

As well as preclinical studies, it would be better if the results of clinical studies are also summarized in table.

Round 2

Reviewer 1 Report

The manuscript entitled “The role of isoflavones in type 2 diabetes prevention and treatment” presents interesting issue, but some areas must be corrected. Unfortunately Author did not take ino account my previous comments.

Major:

There are really important problems associated with the prepared manuscript, including:

  • The serious flaw of the presented manuscript is associated with the fact, that it presents a highly subjective review, not a systematic review. While the systematic review has a key role for broadening knowledge, the other reviews don’t have such role.
  • Taking into account, that the Materials and methods section is not presented (it should be added), without any specific information, it is hard to understand which studies were included into review and why. Authors did not present any key words, which were used during literature search, inclusion and exclusion criteria of references, information about the procedure of literature search conducted by them, number of chosen references, as well as information if some of them were excluded from the review and on the basis of which criteria. As a number of recent publications that are related to the issue were not included, it is a serious problem.
  • Author do not present the current and comprehensive knowledge associated with the issue. It is associated with the fact that they did not include some important issues, while other were included even if they are not so crucial
  • Including the reviews and meta-analysis into own review is also a highly controversial procedure – in many aspects, Author just repeated the conclusions of other authors, without own analysis or conclusions.

Moreover, the manuscript is still shabbily prepared.

Reviewer 2 Report

The authors have responded to the comments appropriately.
